# Hybrid Optimization of Green Supply Chain Network and Scheduling in Distributed 3D Printing Intelligent Factory

Yuran Jin * and Cheng Gao

School of Business Administration, University of Science and Technology Liaoning, Anshan 114051, China
* Correspondence: jinyuran@ustl.edu.cn

**Abstract:** Considering the advantages of 3D printing, intelligent factories and distributed manufacturing, the 3D printing distributed intelligent factory has begun to rise in recent years. However, because the supply chain network of this kind of factory is very complex, coupled with the impact of customized scheduling and environmental constraints on the enterprise, the 3D printing distributed intelligent factory is facing the great challenge of realizing green supply chain networks and optimizing production scheduling at the same time, and thus a theoretical gap appears. This paper studies the hybrid optimization of green supply chain networks and scheduling of the distributed 3D printing intelligent factory. Firstly, according to the green supply chain network architecture of the distributed 3D printing intelligent factory, the cost minimization model is constructed. Secondly, mathematical software is used to solve the model, and the scheduling plan can be worked out. Finally, through the simulation analysis, it is concluded that the influencing factors such as demand, factory size and production capacity complicate the production distribution, and it can be observed that the carbon emission cost has gradually become the main factor affecting the total cost. The study has a reference value for the management decision making of the distributed 3D printing intelligent factory under the background of carbon emissions.

**Keywords:** 3D printing; intelligent factory; distributed production; green supply chain network; production scheduling

## 1. Introduction

With the increasing personalized demand of customers and the shortening of the product life cycle, more and more manufacturing enterprises and government departments are beginning to pay attention to personalized production and flexible production [1–4]. The intelligent factory is an important option to solve these problems. The Industrial 4.0 announced by Germany points out that a new supply chain system can be developed to meet individual needs using the intelligent factory with advanced information and communication technologies, such as the Internet of Things, cloud computing, big data and 3D printing technology [5–8]. In this context, many manufacturing enterprises around the world have carried out the practice of building intelligent factories, such as the Electronic Works Amberg in Germany, Harley-Davidson in the United States, and the Foshan Haier Drum Washing Machine Factory and the Shangpin Home in China [9–12]. By connecting the devices in the intelligent factory through the IoT, enterprises can monitor and share information in the cloud system, adjust the process at any time, quickly respond to the individual needs of customers and improve the flexibility of production [13–15]. In recent years, 3D printing technology has been further integrated with the intelligent factory [16,17], for it makes product customization more possible and flexible [18–20]. In addition, considering that the production of products can be transferred through the distributed 3D printing smart factory, which can avoid the impact of capacity constraints on the production process and achieve high operational efficiency, a distributed 3D printing intelligent factory is beginning to appear on the stage of history.

Although the distributed 3D printing intelligent factory is a new kind of enterprise form, it still depends on the development of the supply chain. Production planning, capacity decision making and outsourcing are all long-term problems, and changes in decision making and demand have a great impact on costs [5,21–24]. Supply chain operation and control problems still need to be solved after decision making, such as short-term planning and production decisions, so we need to design supply chain networks to achieve personalized production in order to reduce costs [25–27]. On the other hand, with the increasingly serious environmental problems in the world, the construction of a green supply chain system is an important means to promote national development [28–30], and reducing carbon emissions has become a factor that must be considered in supply chain design [31–33]. To sum up, in this study, we try to solve the following three problems:

RQ1: what is the green supply chain network architecture of the distributed 3D printing intelligent factory?

RQ2: can a hybrid optimization model of green supply chain networks and scheduling of the distributed 3D printing intelligent factory be modeled to minimize the supply chain cost?

RQ3: what insights can be obtained from the computer simulation analysis of the model?

We will first design a green supply chain network architecture of the distributed 3D printing intelligent factory for general manufacturing enterprises. Then, the hybrid optimization problem of green supply chain networks and scheduling of this distributed 3D printing intelligent factory is modeled and simulated. It is one thing to put forward the architecture of a green supply chain network, but what is more important is to work out the production scheduling plan for 3D printing products by solving the model. We will realize the systematic research from architecture design to simulation experiment, which is a new expansion of related research such as distributed production, 3D printing technology, intelligent factory, green supply chain and production scheduling. On the other hand, the simulation model will become the experimental platform of manufacturing enterprise strategy. Manufacturing enterprises that want to transform intelligent factories or upgrade can use it to simulate and analyze their own development so as to support their development decisions. The results of simulation experiments are of great practical significance to general manufacturing enterprises.

The remainder of the paper is organized as follows: Firstly, the related research is reviewed in the second section. Then, the green supply chain network architecture of the distributed 3D printing intelligent factory is designed. Next, the hybrid optimization model of green supply chain networks and scheduling of the distributed 3D printing intelligent factory is constructed, and the simulation analysis is carried out in Section 5. Finally, the conclusion and prospect are discussed in the last section.

## 2. Literature Review

Aiming at the research on the hybrid optimization of green supply chain networks and scheduling of the distributed 3D printing intelligent factory, scholars only began to pay attention to the research of intelligent factories in recent years. The issues related to 3D printing technology, distributed production, green supply chain and production scheduling have been more in-depth, but research on the distributed 3D printing intelligent factory is relatively scarce.

As for related research on the 3D printing intelligent factory, most scholars study 3D printing technology; for example, Park et al. [34] proved that 3D printing technology can make it possible to produce products with high degrees of freedom and complex shapes, such as manufacturing ceramic cores with high mechanical properties, etc. Zhu and Zhu [35] analyzed that 3D printing technology brings more possibilities to the development of clinical medicine such as surgical medical models and implantable bionic devices. Ma et al. [36] thought that 3D printing technology would have an impact on the traditional supply chain architecture. Xing et al. [37] proposed a 3D printing cloud manufacturing platform, which showed that 3D printing can be monitored remotely and in real time

through the IoT so as to realize intelligent production. Deon et al. [38] reviewed the results of 3D printing drug characterization, which provided a solid foundation for the pharmaceutical industry. There are also many scholars who study intelligent factories; for example, Ivanov et al. [39] proposed a short-term dynamic supply chain model under the environment of the intelligent factory industry for the first time. Afrin et al. [40] designed a multi-objective optimization model to solve the problem of robot work assignment in the intelligent factory. With the proposal of "Made in China 2025", Chinese scholars Tang et al. [41] and Yu [42] put forward the construction framework and scheme of the intelligent factory according to the construction background and current situation of the intelligent factory so as to improve its intelligence. Gong et al. [43] established a model to enable flexible intelligent factories to plan labor and investment decisions reasonably. There is also a small amount of research on the 3D printing intelligent factory. For example, scholars Chung, Kim and Lee [5] pointed out the importance of 3D printing and IoT in intelligent factories and proposed a dynamic supply chain model and production operation plan.

For research on distributed production, Srai et al. [44] pointed out that distributed manufacturing brings changes to manufacturing enterprises, which promotes the development of digitalization and infrastructure. Ding and Jiang [45] studied a variety of distributed production control mechanisms to realize a personalized production system. Ji and Jin [46] built an optimization model of distributed production networks for 3D printing manufacturing enterprises. Yin [47] designed the optimal production scheduling scheme to solve the tire production scheduling in distributed factories. Gong et al. [48] studied the distributed production scheduling problem between different workshops and factories for the first time. Scholars Zhang [49] and Wang et al. [50] innovated an algorithm for solving the distributed production scheduling problem in order to make the workshop flexible. Xin et al. [51] proposed a GSS method based on distributed production scheduling to schedule jobs to reduce the load on critical equipment. Lu et al. [52] designed an iterative greedy algorithm to achieve the goal of the shortest completion time and minimum energy consumption in a distributed flow shop.

For research on green supply chain networks and scheduling, most scholars study green supply chain networks. For example, scholars Ramudhin et al. [53] introduced the model formula of carbon–market sensitivity–green supply chain network design, which provides decision makers with the ability to understand the tradeoff between total logistics costs and reducing the impact of greenhouse gases. Wang et al. [54] studied a supply chain network design problem considering environmental factors, which can be used as an effective tool for green supply chain strategic planning. Elhedhli and Merrick [55] designed a supply chain network considering carbon emissions, which shows that the cost of carbon emissions will change the optimal configuration of the supply chain. Coskun et al. [56] studied the design of a green supply chain network based on consumers' green expectations. There are also many scholars who study production scheduling, such as scholars Koç et al. [57], who designed a facility and vehicle scheduling model between production and shipment, which reasonably allocates the utilization of inbound and outbound vehicles. Paithankar and Chatterjee [58] used a maximum flow algorithm and genetic algorithm to solve the problem of effective resource management and maximum cash flow. Xu [59] pointed out that for flexible production, an efficient scheduling scheme is very important in order to improve the production efficiency and customer satisfaction of the manufacturing workshop. Shao et al. [60] showed that more and more enterprises pay attention to the production scheduling problem of distributed factories and verified the effectiveness of NMA algorithm. Zhang et al. [61] designed an advanced planning and scheduling system for mass flexible manufacturing. There is also a small amount of research on green supply chain networks and scheduling. For example, scholars Tanimizu and Amano [62] proposed a new comprehensive scheduling method for production and transportation problems based on the green supply chain network model to reduce carbon dioxide emissions. Scholar Sinaki et al. [63] proposed a multi-objective programming model, which is used to

integrate production scheduling and environmentally sustainable supply chain networks to obtain optimal satisfaction.

To sum up, scholars' research on the 3D printing intelligent factory is on the rise, and the related topics of intelligent factories were deeply studied from the perspective of the application of 3D printing technology, the business model of intelligent factories and the technological application of intelligent factories. In addition, there are a few pieces of research on the hybrid optimization of green supply chain networks and production scheduling, which mainly focus on the design of green supply chain networks and production scheduling optimization. In view of this, this paper proposes a hybrid optimization model of green supply chain networks and scheduling based on the distributed 3D printing intelligent factory and uses LINGO19.0 software to solve the model, which has not been previously carried out in existing research studies.

## 3. Research Design

For the research on the hybrid optimization of green supply chain networks and scheduling of the distributed 3D printing intelligent factory, we firstly designed its supply chain architecture, including the raw material supplier, accessory supplier, core enterprise, outsourcing enterprise and consumer. Secondly, we constructed a hybrid optimization model of green supply chain networks and scheduling of the distributed 3D printing intelligent factory. The main cost items include fixed cost, production cost, transportation cost, procurement cost and carbon emission cost. After that, a simulation analysis was carried out based on this model to explore the influence of demand, factory size, production capacity and carbon price on the cost. Finally, we draw some conclusions, which can be used as a reference for the development of manufacturing enterprises. The research framework is visually shown in Figure 1.

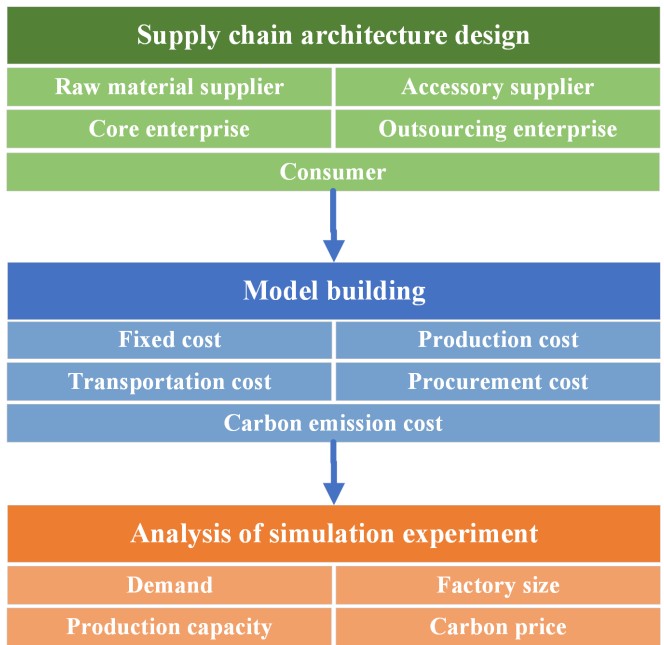

**Figure 1.** Research frame diagram.

## 4. Problem Description and Conditional Hypothesis

### 4.1. Problem Description

The transformation of manufacturing enterprises promotes the upgrading of supply chain structure, and different manufacturing enterprises also have different supply chain structures, so the supply chain architecture of the distributed 3D printing intelligent factory studied in this paper is as follows:

- Raw material supplier: The modern supply chain can design personalized products according to the needs of consumers. After waiting for customers to confirm orders, they can prepare bills of materials and contact raw material suppliers. This can not only respond to customer needs but also greatly reduce inventory and increase enterprise efficiency;
- Accessory supplier: If customers need more complex products, they can not only use 3D printing technology to complete the manufacture of high-degree-of-freedom products but also purchase the necessary accessories from accessories suppliers for assembly and manufacturing. In this way, the time of product manufacturing can be reduced and the production efficiency can be improved;
- Core enterprise: The core enterprise of this paper is the 3D printing intelligent factory, which can realize product personalization, design coordination, supply agility, manufacturing flexibility, service initiative and intelligent decision making. The specific links include intelligent design, intelligent production, intelligent logistics, intelligent products and services [64–66];
- Outsourcing enterprise 3D printing intelligent factory: when the production capacity of a core enterprise is saturated or production stagnates due to failure, the product can be continued to be produced by the outsourcing enterprise. The core enterprise will formulate the product production scheduling plan according to the production capacity of the outsourcing enterprise 3D printing intelligent factory and carry out collaborative manufacturing through information technology and outsourcing enterprises. Additionally, the core enterprise can also monitor the production process of the product in the outsourcing company in real time, which greatly improves the production efficiency of the product and avoids the loss caused by late delivery;
- Consumer: After the production is completed, the product delivery link is carried out, and the supply on demand is realized in the supply chain of the distributed 3D printing intelligent factory, which not only reduces the inventory cost but also achieves the effect of rapid response to customer demand, providing the possibility to meet the individual needs of consumers.

The supply chain of the perfect distributed 3D printing intelligent factory has good coordination so that procurement, design, production and logistics can be combined organically. The resources such as logistics, capital flow and information flow can be distributed rationally so that the upstream and downstream enterprises of the supply chain can achieve common manufacturing goals and win–win cooperation. The supply chain of the distributed 3D printing intelligent factory studied in this paper is shown in Figure 2.

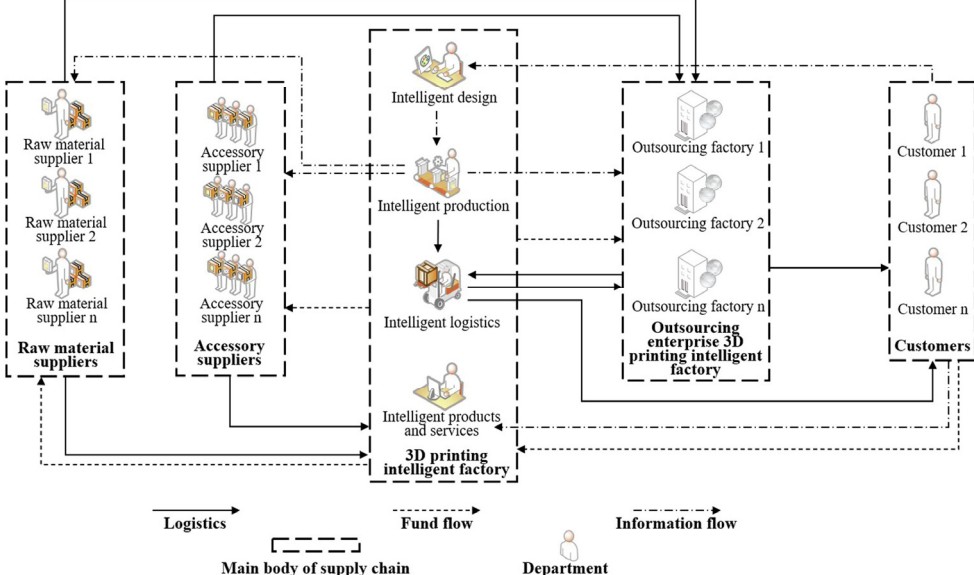

**Figure 2.** Supply chain architecture of distributed 3D printing intelligent factory.

The mixed optimization problem of the green supply chain network and scheduling studied in this paper is based on the distributed 3D printing intelligent factory, choosing the two-level green supply chain network, that is, studying the supplier selection and production process. Through the analysis of the fixed cost, production cost, transportation cost, raw material purchase cost and carbon emission cost of the 3D printing intelligent factory, the production scheduling optimization model of the product is constructed. While minimizing the total cost, it also solves the reasonable scheduling problem between factories, works out the best production plan and realizes a flexible, personalized, automatic, intelligent and efficient production mode.

### 4.2. Assumed Condition

According to the description of the above problems, this paper puts forward the following basic assumptions:

- The quantity of raw materials required for the production of products is consistent with that provided by the supplier;
- The mode of transportation is road transportation, and the weather factors are not taken into the account;
- There is no failure in the operation of the machine;
- The transportation cost between the supplier and the 3D printing intelligent factory is provided by the supplier;
- The carbon emission cost of the core enterprises in the upstream and downstream of the supply chain is mainly considered.

## 5. Model Building

### 5.1. Definition and Description of Symbols

Parameters:

- $I$ is the quantity of the process ($i \in I$, where $i$ is the number of processes);
- $K$ is the quantity of the factory ($k, l \in K$, where $k, l$ is the number of factories);
- $P$ is the quantity of the supplier ($p \in P$, where $p$ is the number of suppliers);
- $D$ is the customer's demand for products;
- $b$ is the quantity of raw materials required per unit of product;
- $AT_{ik}$ is the average daily available time of the process $i$ in the factory $k$ during the planned period;
- $u_{ik}$ is the utilization rate of the process $i$ in the factory $k$;
- $G_k$ is the fixed cost of the selected factory $k$;
- $S_{ik}$ is the installation cost of the process $i$ in the factory $k$;
- $PS_{ik}$ is the process cost per unit of the process $i$ in the factory $k$;
- $PT_{ik}$ is the processing time per unit of process $i$ in the factory $k$;
- $H_{kl}$ is the transportation cost from factory $k$ to factory $l$;
- $A_{ik}$ is the relation matrix between the process $i$ and the factory $k$;
- $YP_p$ is the price of raw materials per unit of product provided by the supplier $p$;
- $CP$ is the price of $CO_2$;
- $QL_{pk}$ is the full life cycle $CO_2$ emissions per unit of raw materials provided by the supplier $p$ to the factory $k$;
- $SL_{ik}$ is the $CO_2$ emission per unit of product produced by the process $i$ in the factory $k$;
- $CL$ is the $CO_2$ emission per unit of transportation process;
- $SD_{pk}$ is the transportation distance from the supplier $p$ to the factory $k$;
- $FD_{kl}$ is the transportation distance from factory $k$ to factory $l$;
- $XS_p$ is the selection cost of supplier $p$;

Decision variable:

- $x_k$ is the choice of factory (1 if factory $k$ is chosen; 0 otherwise);
- $y_{ik}$ is the choice of process (1 if process $i$ in factory $k$ is chosen; 0 otherwise);

- $z_{ikl}$ is the choice of transportation (1 if the product from process $i$ in factory $k$ is sent to factory $l$; 0 otherwise);
- $m_{ik}$ is the production quantity of process $i$ in factory $k$;
- $n_p$ is the choice of supplier (1 if supplier $p$ is chosen; 0 otherwise).

*5.2. Model Building*

We conducted research on the hybrid optimization of green supply chain networks and scheduling in the distributed 3D printing intelligent factory. The main cost items are as follows:

- The fixed cost of the distributed 3D printing intelligent factory.

If the 3D printing intelligent factory $k$ is in operation, the total operating cost of the distributed 3D printing intelligent factory can be obtained by the product of its fixed cost $G_k$ and 0–1 decision variable $x_k$, namely $\sum_{k \in K} G_k x_k$.

- The process installation cost of distributed 3D printing intelligent factory.

If you need to use the process $i$ of 3D printing intelligent factory $k$, the total process installation cost of the distributed 3D printing intelligent factory can be obtained by the product of the installation cost $S_{ik}$ and the 0–1 decision variable $y_{ik}$, namely $\sum_{i \in I} \sum_{k \in K} S_{ik} y_{ik}$.

- The process cost of distributed 3D printing intelligent factory.

If the process $i$ of 3D printing intelligent factory $k$ is used to produce products, the total process cost of the distributed 3D printing intelligent factory can be obtained by the product of its unit process cost $PS_{ik}$ and its production quantity $m_{ik}$, namely $\sum_{i \in I} \sum_{k \in K} PS_{ik} m_{ik}$.

- The transportation cost between distributed 3D printing intelligent factories.

If the process $i$ needs to be sent from distributed 3D printing intelligent factory $k$ to factory $l$, the total processing time can be obtained by the product of each unit of processing time $PT_{ik}$ and its production quantity $m_{ik}$ and then divided by the average daily available time $AT_{ik}$ during the planning period and the proportion of daily logistics cost occupied by process $i$ can be obtained. Then, the total transportation cost between distributed 3D printing intelligent factories can be obtained by multiplying the daily transportation cost $H_{kl}$ and 0–1 decision variable $z_{ikl}$, namely $\sum_{i \in I} \sum_{k \in K} \sum_{l \in K} \frac{PT_{ik} m_{ik}}{AT_{ik}} H_{kl} z_{ikl}$.

- The supplier selection cost of the distributed 3D printing intelligent factory.

If the supplier $p$ is selected in the distributed 3D printing intelligent factory, the supplier selection cost of the distributed 3D printing intelligent factory can be obtained by the product of the supplier selection cost $XS_p$ and the 0–1 decision variable $n_p$, namely $\sum_{p \in P} XS_p n_p$.

- The price of raw materials provided by the supplier to the distributed 3D printing intelligent factory.

If the process $i$ of 3D printing intelligent factory $k$ chooses supplier $p$, and each unit of product needs $b$ units of raw materials, the total raw material price provided by the supplier to the distributed 3D printing intelligent factory can be obtained by multiplying the raw material price $YP_p$ provided by the supplier, the supplier's choice $n_p$ and the product quantity m of 3D printing intelligent factory process 1, namely $\sum_{p \in P} \sum_{i=1} \sum_{k \in K} b \cdot YP_p m_{ik} n_p$.

- The $CO_2$ emission cost of the raw materials provided by the supplier to the distributed 3D printing intelligent factory throughout its life cycle.

If the process $i$ of 3D printing intelligent factory $k$ chooses supplier $p$, the total life cycle $CO_2$ emissions of raw materials supplied by supplier $p$ to 3D printing intelligent factory $k$ can be obtained by multiplying the total $CO_2$ emissions of raw materials provided by suppliers to distributed 3D printing intelligent factories through the product of $QL_{pk}$ in the whole life cycle of raw materials and $m_{1k}$ in process 1. Then, multiplied by the price of $CO_2$ $CP$, the total $CO_2$ emission cost of the raw materials provided by the supplier to the

distributed 3D printing intelligent factory in the whole life cycle can be obtained, namely $\sum_{p\in P}\sum_{i\in I}\sum_{k\in K}b\cdot QL_{pk}m_{ik}CP\cdot n_p$.

- The $CO_2$ emission cost during transportation from the supplier to the distributed 3D printing intelligent factory.

If the process $i$ of 3D printing intelligent factory $k$ chooses supplier $p$, the total $CO_2$ emissions per kilometer during transportation can be obtained by multiplying the product m and the $CL$ generated per kilometer during transportation, and then, multiplied by the transportation distance from supplier $p$ to 3D printing intelligent factory $k$ $SD_{pk}$, the $CO_2$ emissions during transportation from the supplier to the 3D printing intelligent factory can be obtained, which is then multiplied by the price $CP$ of $CO_2$. The total $CO_2$ emission cost of the supplier's transportation to the 3D printing intelligent factory can be obtained, namely $\sum_{p\in P}\sum_{i\in I}\sum_{k\in K}CP\cdot SD_{pk}CL\cdot m_{ik}n_p$.

- The $CO_2$ emission cost of products produced by the distributed 3D printing intelligent factory.

If the process $i$ produces the product in the 3D printing intelligent factory $k$, the product of the $CO_2$ emission $SL_{ik}$ generated during its production and the product quantity m can be obtained by multiplying the $CO_2$ emission generated by the 3D printing intelligent factory product and then multiplied by the $CO_2$ price $CP$ to obtained the total $CO_2$ emission cost generated by the 3D printing intelligent factory product, namely $\sum_{i\in I}\sum_{k\in K}CP\cdot SL_{ik}m_{ik}$.

- The $CO_2$ emission cost during transportation between distributed 3D printing intelligent factories.

If process $i$ needs to be sent from 3D printing intelligent factory $k$ to factory $l$, the total processing time can be obtained by the product of each unit of processing time $PT_{ik}$ and its production quantity $m_{ik}$ and then divided by the average daily available time $AT_{ik}$ during the planned period to obtain the number of products that can be produced. Then, multiplying by the product of the $CO_2$ emissions per kilometer $CL$ and the 0–1 decision variable $z_{ikl}$, the 3D printing of the number of products transported between intelligent factories can be obtained, and multiplying by the $CO_2$ emissions per kilometer generated during transportation $CL$, the $CO_2$ emissions of 3D printing intelligent factories during transportation can be obtained. Then, the transportation distance between 3D printing intelligent factories $FD_{kl}$ is multiplied to obtain the total $CO_2$ emissions during transportation between 3D printing intelligent factories, and then it is multiplied by the price of $CO_2$ to obtain the total $CO_2$ emission cost between 3D printing intelligent factories during transportation, namely $\sum_{i\epsilon I}\sum_{k\epsilon K}\sum_{l\epsilon K}CP\frac{PT_{ik}m_{ik}}{AT_{ik}}z_{ikl}FD_{kl}CL$.

Through the above analysis, the objective function of the green supply chain network optimization problem of the distributed 3D printing intelligent factory is as follows:

$$
\begin{aligned}
\text{Min}\Big( &\sum_{k\in K}G_kx_k + \sum_{i\in I}\sum_{k\in K}S_{ik}y_{ik} + \sum_{i\in I}\sum_{k\in K}PS_{ik}m_{ik} + \sum_{i\in I}\sum_{k\in K}\sum_{l\in K}\frac{PT_{ik}m_{ik}}{AT_{ik}}H_{kl}z_{ikl} \\
&+ \sum_{p\in P}XS_pn_p + \sum_{p\in P}\sum_{i=1}\sum_{k\in K}b\cdot YP_pm_{ik}n_p \\
&+ \sum_{p\in P}\sum_{i=1}\sum_{k\in K}b\cdot QL_{pk}m_{ik}CP\cdot n_p + \sum_{p\in P}\sum_{i\in I}\sum_{k\in K}CP\cdot SD_{pk}CL\cdot m_{ik}n_p \\
&+ \sum_{i\in I}\sum_{k\in K}CP\cdot SL_{ik}m_{ik} + \sum_{i\epsilon I}\sum_{k\epsilon K}\sum_{l\epsilon K}CP\cdot\frac{PT_{ik}m_{ik}}{AT_{ik}}z_{ikl}FD_{kl}CL\Big)
\end{aligned}
\tag{1}
$$

$$s.t. \begin{cases} m_{ik} \leq D \cdot y_{ik}, & \forall i,k \\ y_{ik} \leq A_{ik}, & \forall i,k \\ \sum_{k \in K} m_{ik} = D, & \forall i \\ \sum_{i \in I} s_{ik}/I \leq x_k, & \forall k \\ PT_{ik}m_{ik} \leq u_{ik}AT_{ik}y_{ik}, & \forall i,k \\ z_{ikl} \leq y_{ik}, & \forall i,k,l \\ z_{ikl} \leq y_{i+1\,l}, & \forall i,k,l \\ z_{ikl} \geq y_{ik} + y_{i+1\,l} - 1, & \forall i,k,l \\ \sum_{p \in P} n_p \geq 1 \\ m_{ik} \geq 0, & \forall i,k \\ x_k, y_{ik}, z_{ikl} \in \{0,1\}, & \forall i,k,l \end{cases} \quad (2)$$

## 6. Numerical Experiment and Result Analysis

*6.1. Hybrid Optimization Analysis of Green Supply Chain Network and Scheduling of Distributed 3D Printing Intelligent Factory*

6.1.1. Data Description

Suppose that Company A is a manufacturer of 3D printing products, with a factory with four core production processes. In this paper, the parameter data used in the simulation experiment are randomly set according to the actual situation, and the data range of each parameter is shown in Table 1, where $H_{kl}$ and $FD_{kl}$ are diagonal symmetric matrices with diagonal 0. Three-dimensional printers will take a long time in the processing stage, including modeling, fumigation and so on. Suppose the planned time is 5 days and the factory operates 8 h a day. In addition, transportation between factories takes place at the end of 8 h of working hours. The solving software used in this paper is LINGO19.0, and simulation experiments are carried out on a 2.40 GHz 4-core Intel Core i5 processor and 16GB computer.

**Table 1.** Data range of each parameter.

| $AT_{ik}$ (h) | [32, 40] | $PT_{ik}$ (h) | [0.5, 1.5] |
|---|---|---|---|
| $u_{ik}$ | [0.8, 1] | $H_{kl}$ (USD) | [0, 5000] |
| $G_k$ (USD) | [5000, 10,000] | $A_{ik}$ | [0, 1] |
| $S_{ik}$ (USD) | [1000, 2000] | $YP_p$ (USD per piece) | [50, 100] |
| $PS_{ik}$ (USD) | [30, 50] | $XS_p$ (USD) | [100, 300] |
| $QL_{pk}$ (ton) | [0.45, 0.5] | $FD_{kl}$ (kilometer) | [0, 100] |

In addition, in the hybrid optimization model of the green supply chain network and scheduling of distributed 3D printing intelligent factories, we consider three raw material suppliers, six distributed 3D printing intelligent factories and a product that requires four processes, namely $p = 3$, $i = 4$ and $k = 6$. In addition, we assume that the price of $CO_2$ is 50 USD per ton and that the $CO_2$ emission per unit of transportation is 0.005 tons per km, that is, $CP = 50$ and $CL = 0.005$.

6.1.2. Experimental Results

Table 2 shows the relationship matrix between 6 distributed 3D printing intelligent factories and their 4 process sizes, where 1 represents the operational process and 0 represents the non-operational process. For example, when an order is generated, 3D Printing Intelligent Factory 1 has one, three and four processes that can be run.

**Table 2.** Data matrix of parameter $A_{ik}$.

| $A_{ik}$ | | Factory | | | | | |
|---|---|---|---|---|---|---|---|
| | | 1 | 2 | 3 | 4 | 5 | 6 |
| Process | 1 | 1 | 0 | 0 | 0 | 1 | 0 |
| | 2 | 0 | 1 | 0 | 1 | 0 | 0 |
| | 3 | 1 | 1 | 0 | 1 | 1 | 0 |
| | 4 | 1 | 0 | 1 | 1 | 0 | 0 |

In the green supply chain network and scheduling hybrid optimization model of the distributed 3D printing intelligent factory, we assume that each product needs 5 raw materials and the demand is 50, that is, $b = 5$ and $D = 50$; the running result of the model is shown in Figure 3, where the solid square represents the operable process and the dotted square represents the non-operable process. Here, we can see that supplier 1 is selected for raw material supply. First, raw materials are transported from supplier 1 to factory 1 for the production of process 1. Due to the limited production capacity, the products need to be transported to factory 2 and factory 4 for the production of the second process, and factory 1 and factory 4 are selected for the third process production. The fourth process selects factory 1, factory 3 and factory 4, with a total cost of USD 99593.74 (here, the two-stage method is used to further optimize the transportation cost and carbon emission cost in the process of transportation. The priority transportation in the same factory and close distance can reduce the cost of the process of transportation).

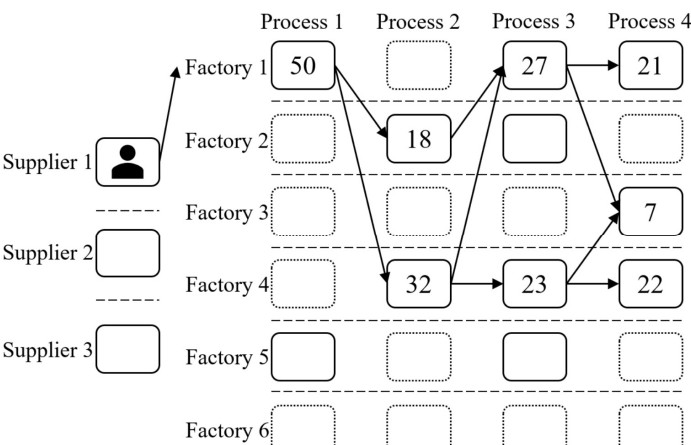

**Figure 3.** Green supply chain network of distributed 3D printing intelligent factory with demand of 50.

After designing the green supply chain network of the 3D printing intelligent factory, the production of the product can be scheduled using the results of the operation model, as shown in Table 3. In the case of a demand of 50, if a process of a product needs to be produced in an hour, then if you work 8 h a day, you can produce 8 products a day, so the first process can produce 8 products on the first day of the first factory. Since the transportation occurs only after the end of the day's work, the products will be shipped to factory 2 for the production of the second process the next day, process 1 of factory 1 on the same day can produce 8 products, and so on. The customer will receive the product after ten days.

Combined with the experimental results, we can observe that the main factors affecting the green supply chain network and total cost of distributed 3D printing intelligent factories may be demand, factory size, production capacity and carbon price. Then, we will analyze the impact on the cost through these four factors.

**Table 3.** A scheduling scheme with a demand of 50.

| Process (Factory) | Quantity of Products to Be Produced (pcs) | | | | | | | | | |
|---|---|---|---|---|---|---|---|---|---|---|
| | 1st day | 2nd day | 3rd day | 4th day | 5th day | 6th day | 7th day | 8th day | 9th day | 10th day |
| 1(1) | 8 | 8 | 8 | 8 | 8 | 8 | 2 | 0 | 0 | 0 |
| 2(2) | 0 | 8 | 8 | 2 | 0 | 0 | 0 | 0 | 0 | 0 |
| 2(4) | 0 | 0 | 0 | 6 | 8 | 8 | 8 | 2 | 0 | 0 |
| 3(1) | 0 | 0 | 8 | 8 | 2 | 0 | 0 | 7 | 2 | 0 |
| 3(4) | 0 | 0 | 0 | 6 | 8 | 8 | 2 | 0 | 0 | 0 |
| 4(1) | 0 | 0 | 8 | 9 | 3 | 0 | 0 | 2 | 0 | 0 |
| 4(3) | 0 | 0 | 0 | 0 | 0 | 0 | 0 | 2 | 5 | 2 |
| 4(4) | 0 | 0 | 0 | 6 | 7 | 9 | 1 | 0 | 0 | 0 |

*6.2. Sensitivity Analysis*

6.2.1. The Influence of Demand on Cost

Demand is a very important factor in the green supply chain network of the distributed 3D printing intelligent factory. We will analyze the impact of various costs from demand 1, demand 10, demand 20, demand 30, demand 40 and demand 50. First of all, the supply chain results of demand 1, demand 20, demand 30 and demand 40 are displayed, as shown in Figure 4. Here, supplier 1 is still selected to supply raw materials. According to the results of their comparison, different supply chains can be obtained when other parameters are the same and only the demand is different, which shows that demand has a great impact on the network structure of the green supply chain. The resulting scheduling results are obviously different, and it is found that when the demand is between 1 and 20, the selected factories are the same, and when it reaches more than 20, the number of selected factories only increases.

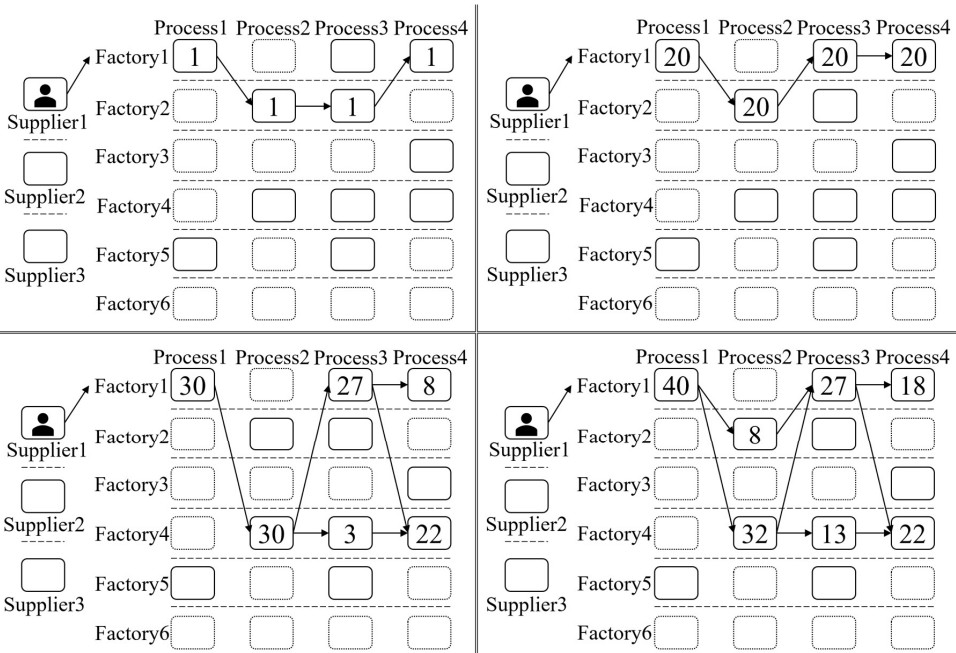

**Figure 4.** Green supply chain operation of distributed 3D printing intelligent factory with demand of 1, 20, 30 and 40.

Figure 5 shows that the proportion of each cost is connected to the change in demand. When the demand is from 1 to 20, the numbers of factories selected are the same, so the fixed costs are the same, and when the demand is more than 20, as more and more factories are selected, the growth rate of fixed costs become larger and larger, and transportation costs change differently due to the choice of different routes. The production cost has less

and less influence on the total cost with the increase in demand, and the proportion of carbon emission cost is relatively small when the demand is from 1 to 10, indicating that it has little influence on the total cost. When the demand is more than 20, it has a greater impact on the total cost. In addition, as a whole, it can be found that with the growth of demand, the impact of fixed cost on the total cost becomes smaller and smaller, and the cost of carbon emissions becomes the main factor affecting the total cost.

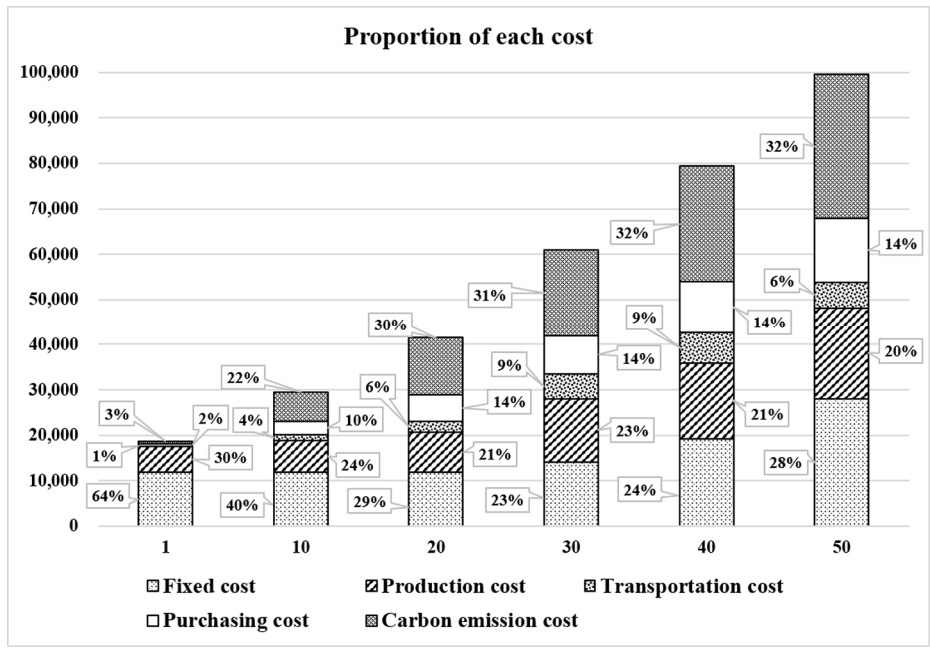

**Figure 5.** Compare the costs according to the changes in demand.

6.2.2. The Influence of Factory Size on Cost

Next, we will observe the changes in costs by changing the size of the factory (that is, the number of factories). Here, we fix the demand at 40 and the number of processes at 4 and set the factory size to 4, 5, 6, 7, 8 and 9. Figure 6 visually shows the green supply chain network operation of 5, 6, 7 and 8 factories, reflecting the impact of the increase in factory size on production distribution. Factors such as the production capacity of a new factory or unit transportation cost are all important factors that may lead to the re-selection of production distribution, so the increase in production scale may lead to different green supply chain network operations. It is not necessarily that the larger the production scale, the more factories you choose.

Next, we can sum up the costs of each link and obtain the proportion chart shown in Figure 7. It can be seen that with the increase in the factory size, the total cost does not change greatly, and when the factory size is 6, the total cost is relatively minimal. This is because with the re-selection of production distribution, fixed costs and transportation costs will fluctuate greatly. On the other hand, the proportion of production cost is relatively stable, the proportion of transportation cost is decreasing and the proportion of carbon emission cost is increasing as a whole, and it can be found that when the factory scale is 6, 8 and 9, only 3 factories are selected to produce; at this time, the proportion of carbon emission cost is more than 30%, which shows that with the increase in factory size, the impact of carbon emission cost on the total cost becomes greater and greater. The fewer the number of factories selected, the greater the impact of carbon emission costs on the total cost.

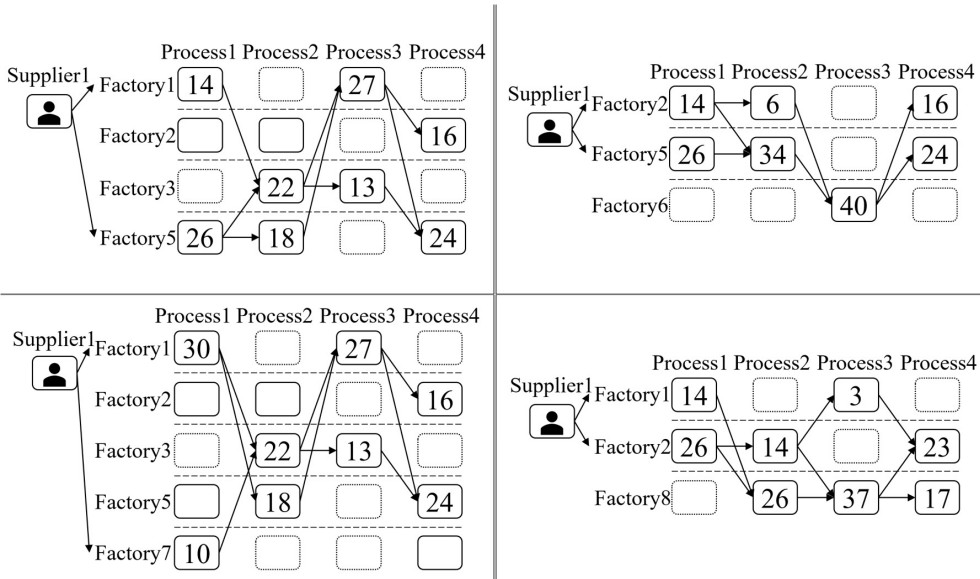

**Figure 6.** Green supply chain operation of distributed 3D printing intelligent factory with factory scale of 5, 6, 7 and 8 factories.

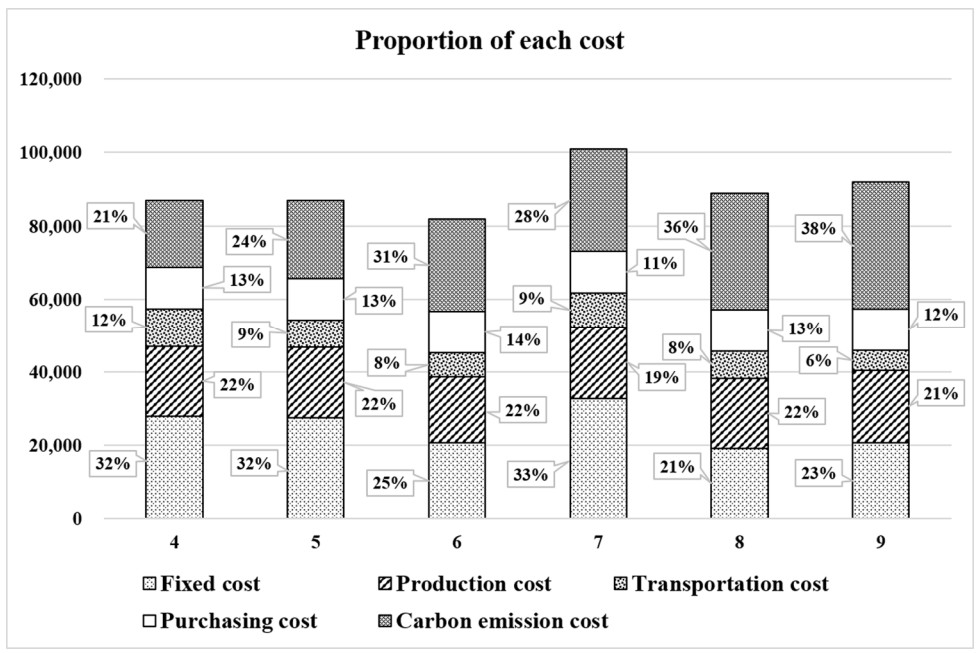

**Figure 7.** Compare the costs according to the changes in the size of the factory.

6.2.3. The Influence of Production Capacity on Cost

Next, we still set the production scale at 6 factories and 4 processes, and the demand is set at 40 to observe the impact of production capacity on costs:

1.  The influence of the change in the overall production capacity of the factory on the cost.

First of all, we compare and analyze the overall production capacity of the factory by reducing it by 20%, reducing it by 10%, increasing it by 10%, increasing it by 20% and increasing it by 30% and observe the impact on costs. As shown in Figure 8, showing the green supply chain network of distributed 3D printing intelligent factories with reduced production capacity by 20% and 10% and increased production capacity by 10% and 30%, it can be seen that if production capacity becomes smaller, some processes may require more factories, complicating the green supply chain and increasing total costs. It can also be seen

that the design of a good green supply chain structure needs to be combined with real-time data, including demand, factory size and production capacity.

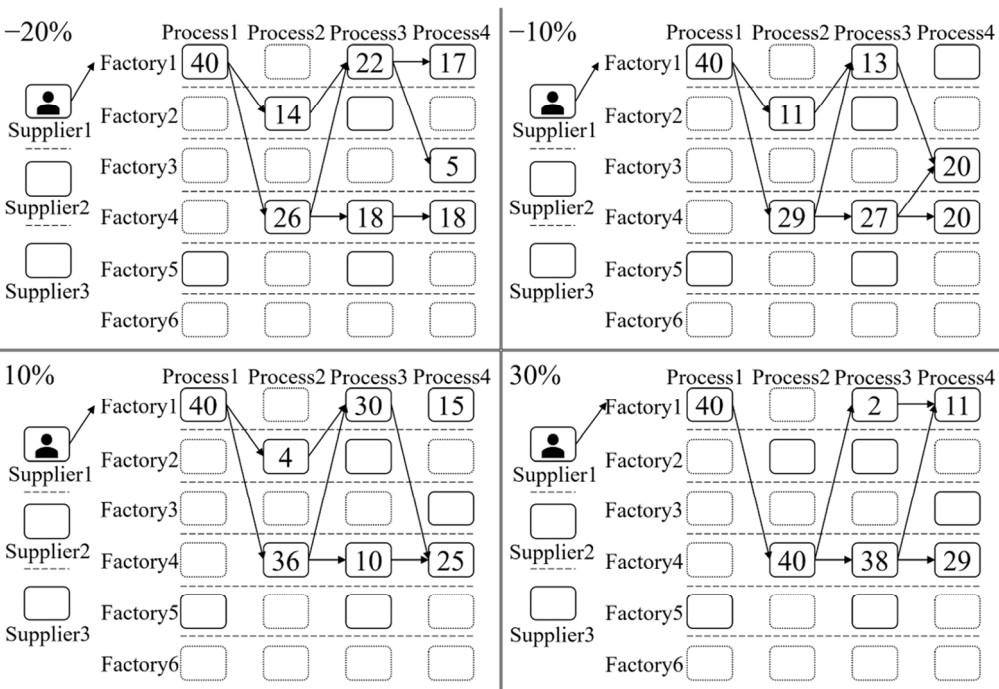

**Figure 8.** Production capacity reduced by 20%, reduced by 10%, increased by 10% and increased by 10% in the green supply chain network operation of distributed 3D printing intelligent factory.

Next, when we sum up the costs of each link, we can obtain the proportion chart shown in Figure 9, which shows that because the number of needed factories will become smaller with the increase in production capacity, both fixed costs and production costs will show a downward trend, while transportation costs will rise briefly under low production capacity and then fall sharply and become smaller and smaller. In addition, with the increase in production capacity, the impact of fixed cost and transportation cost on the total cost becomes smaller and smaller, while the impact of procurement cost and carbon emission cost on the total cost increases.

2.  The influence of the change in production capacity of a single factory on cost.

Next, after screening, we compared the production capacity of factory 1 and factory 4 by reducing it by 20%, reducing it by 10%, increasing it by 10%, increasing it by 20% and increasing it by 30% and observed the impact on costs. As shown in Figure 10, the production capacity of 3D printing intelligent factory 1 and factory 4 was reduced by 20% and increased by 20% in the green supply chain network operation, respectively. It can be seen that a change in the production capacity of a single plant can also lead to a significant change in production distribution, which needs to be combined with real-time data.

Figure 11 shows the impact of the production capacity of different factories on the total cost. It can be found that the production capacity of factory 1 is relatively stable for the total cost after −10%, while the production capacity of factory 4 as a whole leads to a downward trend in the total cost. This shows that when the production capacity is not particularly small, the change in the production capacity of factory 4 has a greater impact on the total cost, while the change in the production capacity of factory 1 has little impact on the total cost. Therefore, if the objective is to reduce the total cost, one can consider mainly increasing the production capacity of 3D printing intelligent factory 4, which also shows that the increase in the production capacity of factories with continuous runnable processes has the greatest impact on the total cost.

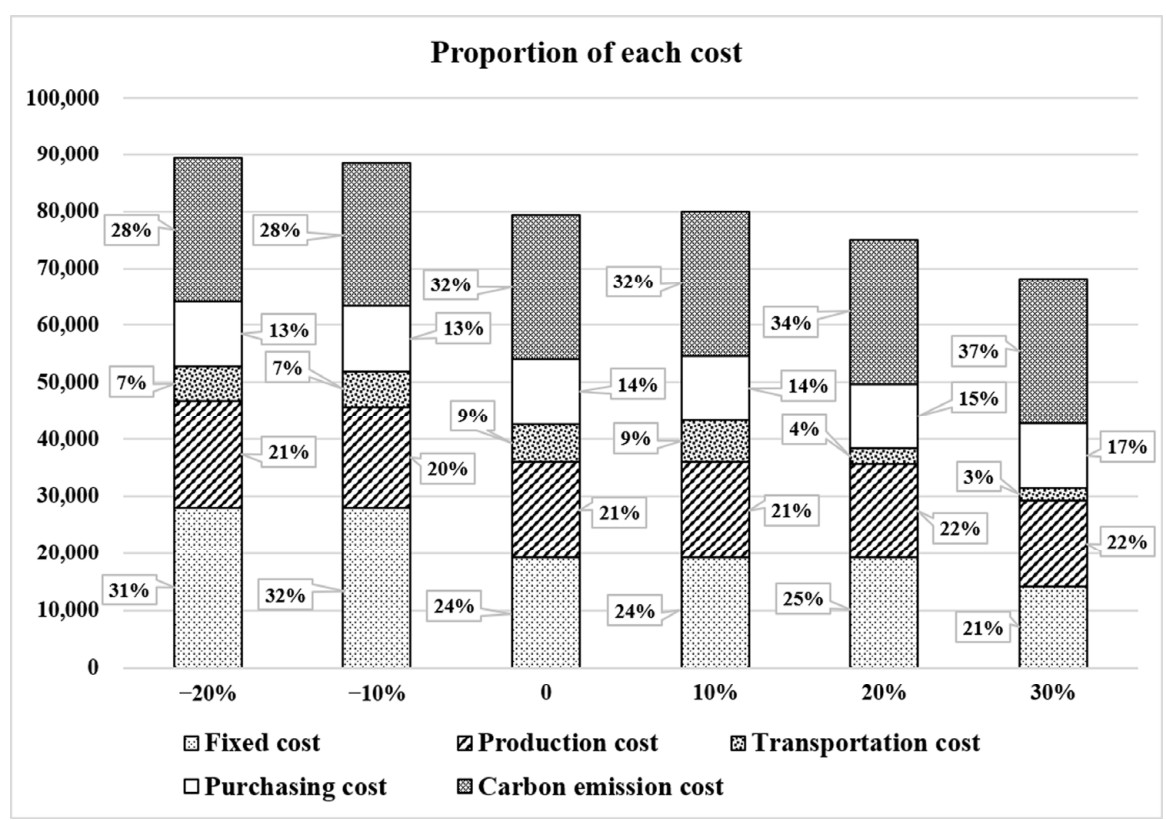

**Figure 9.** Compare the costs according to the changes in production capacity.

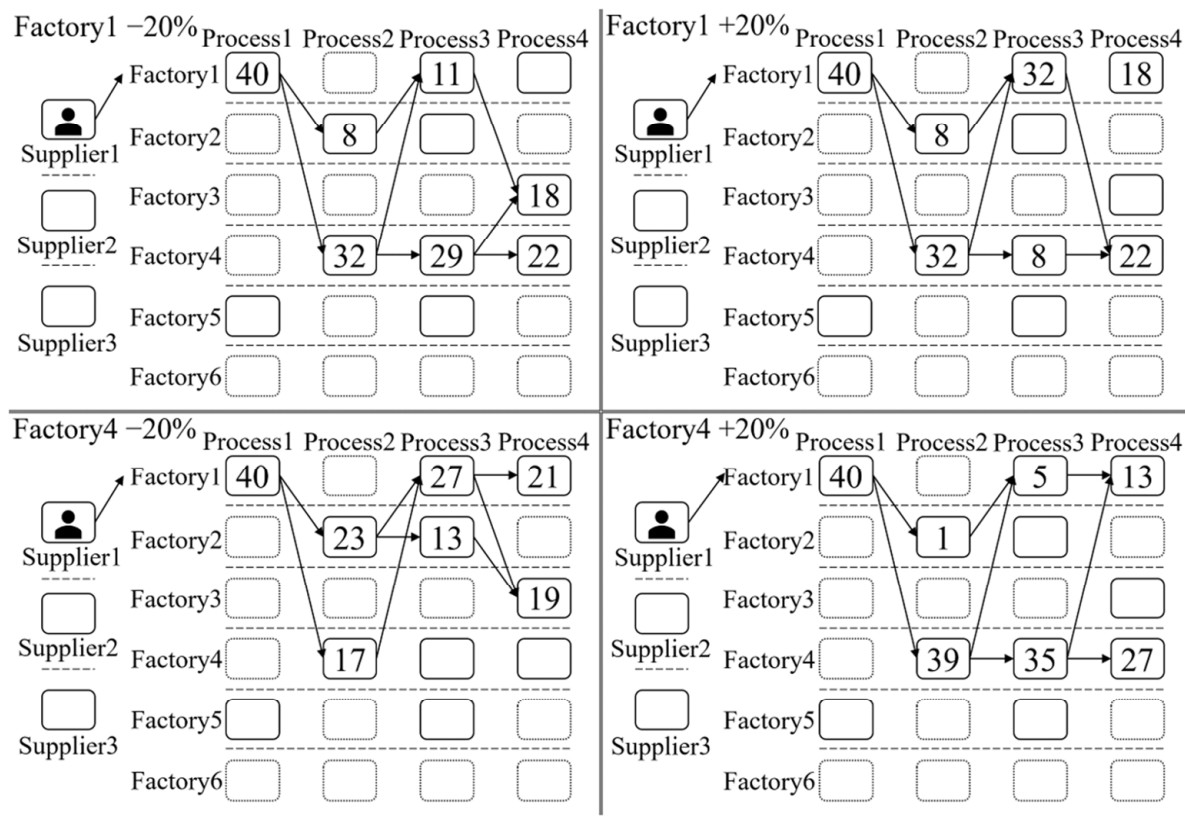

**Figure 10.** The production capacity of 3D printing intelligent factory 1 and factory 4 reduced by 20% and increased by 20% in green supply chain network operation.

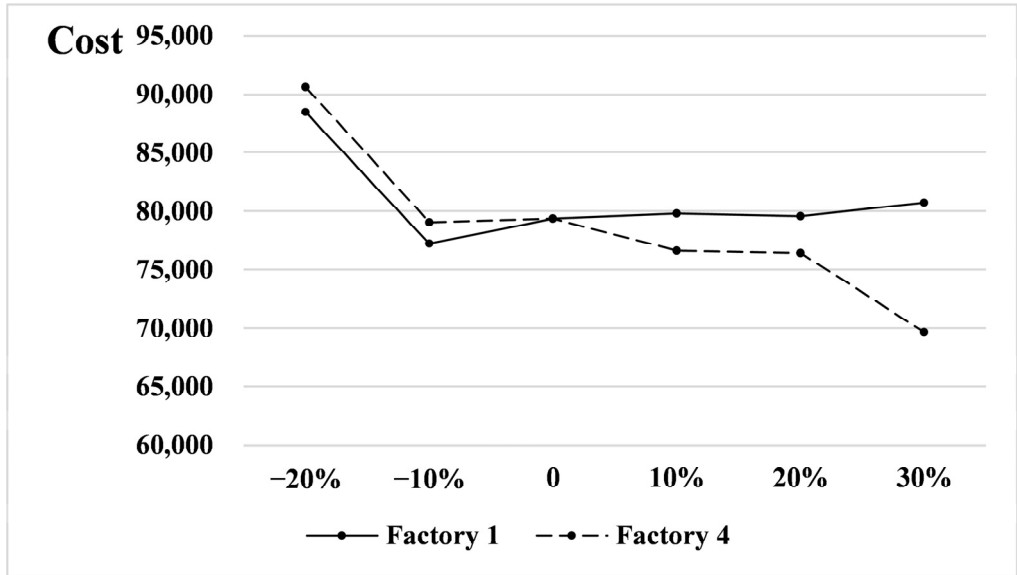

**Figure 11.** The influence of production capacity of different factories on total cost.

6.2.4. The Influence of Carbon Price on Cost

In the hybrid optimization model of green supply chain networks and scheduling of the distributed 3D printing intelligent factory, we will discuss its impact on the total cost by setting different carbon prices and observe the impact on the total cost according to different demand, as shown in Figure 12, we can see that the total cost increases greatly with the increase in demand at high carbon prices; conversely, the increase in total cost is small at low carbon prices. On the other hand, when the demand is low, the increase in the price of carbon will not increase the total cost greatly; on the contrary, when the demand is large, the total cost will increase significantly. This shows that with the growth of demand, the impact of carbon price on the total cost of manufacturing enterprises has become increasingly prominent. In the case of increasing carbon price, enterprises need to continuously optimize and control carbon emissions.

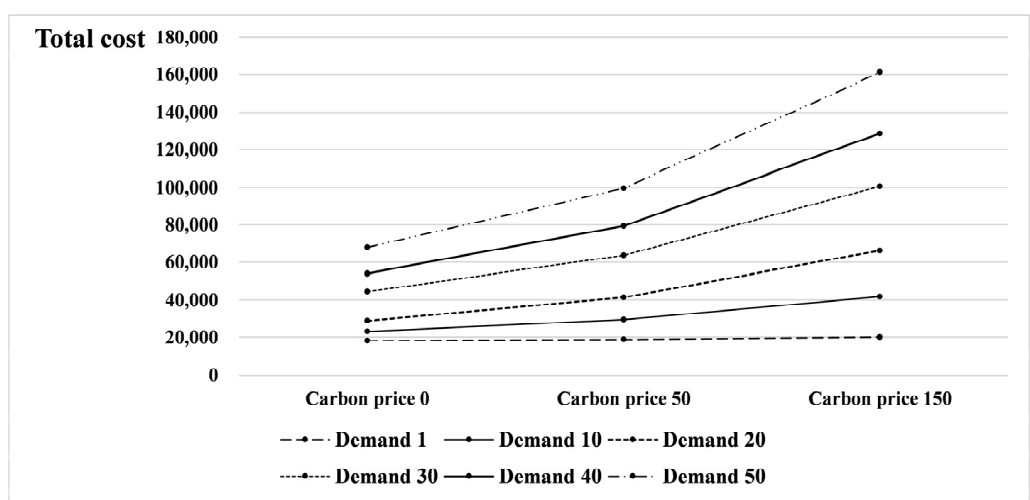

**Figure 12.** The influence of carbon price on total cost under different demand.

**7. Conclusions and Prospect**

By consulting the relevant references, this paper has reviewed the research on the hybrid optimization of green supply chain networks and scheduling of the distributed 3D printing intelligent factory and constructed the planning model of cost minimization. A mathematical solution software was used to solve the model, and the corresponding

scheduling scheme was worked out. Finally, some conclusions have been drawn through sensitivity analysis, which can be used as a reference for manufacturing enterprises.

### 7.1. Conclusions

In this paper, the hybrid optimization of green supply chain networks and scheduling in the distributed 3D printing intelligent factory is studied, the corresponding cost minimization goal programming model is constructed, the two-stage method is used to schedule the results and the sensitivity analysis is carried out. The simulation results verify the effectiveness of the model proposed in this paper and show that the changes in demand, factory size, overall production capacity and individual production capacity may lead to different production distribution results of the green supply chain network of the distributed 3D printing intelligent factory. Among them, the higher the demand or the lower the production capacity, the more complex the green supply chain structure will be. With the increase in demand, factory size or production capacity, carbon emission cost gradually becomes the main factor affecting the total cost; only when the factory size increases, the fewer factories selected, the greater the impact of carbon emission cost on the total cost. The total cost will decrease with the increase in production capacity, and the increase in the production capacity of factories with continuous operational processes has the greatest impact on the total cost. In addition, with the growth of demand, the impact of carbon price on the total cost of manufacturing enterprises has become increasingly prominent.

### 7.2. Theoretical Implications

The hybrid optimization model of green supply chain networks and scheduling of the distributed 3D printing intelligent factory is proposed for the first time, and the effectiveness of the model is verified by simulation experiments. It is one thing to put forward the architecture of a green supply chain network, but what is more important is to work out the production scheduling plan for 3D printing products by solving the model. Compared with a previous study [5], this study considers the selection of suppliers more, increases the level of the supply chain and considers the factors of carbon emissions; at the same time, it also refines the model and constraints. This study realizes system research from establishment to verification of the hybrid optimization model of green supply chain networks and scheduling based on the distributed 3D printing intelligent factory. The above research is a new expansion of distributed production, 3D printing technology, intelligent factory, green supply chain and production scheduling and has important reference value for related research fields.

### 7.3. Practical Implications

Based on the background of the distributed 3D printing intelligent factory, the hybrid optimization model of green supply chain networks and scheduling proposed for general manufacturing enterprises adapts to the development trend of distributed production, 3D printing intelligent factory and green supply chain. It can provide a valuable reference for manufacturing enterprises that intend to transform and upgrade in order to improve their competitiveness. The green supply chain network and scheduling hybrid optimization model of the distributed 3D printing intelligent factory studied in this paper can be applied to many fields, such as 3D printing clothing, 3D printing cars, 3D printing medical devices and so on. In the future, there will be more and more such enterprises, so the model studied in this paper can be widely used for reference.

From the simulation results, the effects of demand, factory scale, production capacity and carbon price on the cost reflect that the research model has a good enterprise strategy experimental function. Manufacturing enterprises that want to transform 3D printing intelligent factories can use this model to simulate and analyze according to their own development so as to provide an important basis for enterprise development decision making.

### 7.4. Limitations and Future Research

The green supply chain network and scheduling hybrid optimization model of the distributed 3D printing intelligent factory studied in this paper also needs to consider various uncertainties in the actual production environment, such as compatibility, product quality and so on, in order to enrich the model and constraints. In addition, due to the limitation of personal knowledge level and ability, the two-stage method is used to solve the optimal solution. In order to improve the efficiency of the solution, the model needs to be further optimized in the future so that the scheduling scheme can be obtained directly. It can not only improve the speed of the solution but also avoid possible operational errors in the process.

**Author Contributions:** Conceptualization, C.G. and Y.J.; methodology, C.G. and Y.J.; software, C.G.; validation, Y.J.; formal analysis, C.G.; resources, Y.J.; data curation, C.G. and Y.J.; writing—original draft preparation, C.G. and Y.J.; writing—review and editing, C.G. and Y.J.; supervision, Y.J. All authors have read and agreed to the published version of the manuscript.

**Funding:** This work was supported by Social Science Foundation of Liaoning Province (grant number L22BJY040).

**Institutional Review Board Statement:** Not applicable.

**Data Availability Statement:** The original data of this study are assumed by the author according to the actual situation.

**Acknowledgments:** We would like to thank Yu Xue from Lingo China for her great help in solving the problems in this paper and Zhiwei Huang for his guidance on the application of the software.

**Conflicts of Interest:** The authors declare no conflict of interest.

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
