# Peer review of "Hybrid Optimization of Green Supply Chain Network and Scheduling in Distributed 3D Printing Intelligent Factory"

_sustainability, doi:10.3390/su15075948_

Round 1
Reviewer 1 Report
This paper focuses on the green supply chain network and scheduling hybrid optimization model of distributed 3D printing intelligent factory,which has a good framework and relatively complete content. It is a very interesting topic. But the article still has the following parts, which can be improved.
1. The main research content of this paper is "green supply chain network and scheduling". It is suggested to add a form to summarize the literature about this topic.
2. The description of the problems in the abstract is not clear enough.
3. Some variables in the article are lack of clear definition, such as Hkl,mik.
4. Result analysis is not well organized.
5. The quality of English needs improving. We strongly suggest that you obtain assistance from a colleague who is well-versed in English or whose native language is English.
Once the above concerns are fully addressed, the manuscript could be accepted for publication in this journal.
Reviewer 2 Report
Dear authors, the topic of this paper is very interesting now and in the following years, but I recommend you improve some points like:
1º Authors have to check citations on references and paragraphs according to Sustainability' rules, please. For example, Ivanov, et al. [27] is not in Italics.
2º The title is: Research on Hybrid Optimization of Green Supply Chain Network and Scheduling in Distributed 3D Printing Intelligent Factory. Authors need to be more specific in the title, this title is very general, for example, in which sector, technology, tourism, logistics...... A title more short and specific.
3º Keywords should include the specific sector where this research is focused as I previously mentioned.
4º. Congratulations, because authors added updated studies and the entire paper shows a good structure, information, and English grammar.
5º. Authors need to explain why they have written this paper, displays the main goals and research questions, write "To fill this gap...." and they have to give information from updated studies related to this topic. Readers and researchers need to know why this research, and the last information in this topic. I did not see this details in the introduction section.
6º. Literature review section must be improved and being structured according to keywords. For example:
a) 3D printing intelligent factory. Show different studies and examples. Add some images to stage keywords.
b) green supply chain network; c) cost minimization in the sector which authors are speaking...
7º. I do not know where this study was done, why? Authors have to show a map where this research was done. Readers need to know it. It is very important know the localisation, so people will speak about this paper and companies showed.
8º Authors say: Park, et al. [32] proved that 3D printing 66 technology can make it possible to produce products with high degrees of freedom and 67 complex shapes. Can authors show some examples of this?
9º I recommend authors add these authors in the literature review section and results to support their information:
Mai, J., Zhang, L., Tao, F., & Ren, L. (2016). Customized production based on distributed 3D printing services in cloud manufacturing. The International Journal of Advanced Manufacturing Technology, 84(1), 71-83. https://doi.org/10.1007/s00170-015-7871-y
Darwish, L. R., El-Wakad, M. T., & Farag, M. M. (2021). Towards sustainable industry 4.0: A green real-time IIoT multitask scheduling architecture for distributed 3D printing services. Journal of Manufacturing Systems, 61, 196-209. https://doi.org/10.1016/j.jmsy.2021.09.004
Srai, J. S., Kumar, M., Graham, G., Phillips, W., Tooze, J., Ford, S., ... & Tiwari, A. (2016). Distributed manufacturing: scope, challenges and opportunities. International Journal of Production Research, 54(23), 6917-6935. https://doi.org/10.1080/00207543.2016.1192302
Florido-Benítez, L. and Aldeanueva, I. F. (2022). Fusing International Business and Marketing: A bibliometric study. Administrative Sciences, 12(4), p. 159. https://doi.org/10.3390/admsci12040159
10º. Authors write: According to the description of the above problems, this paper puts forward the 179 following basic assumptions in the lines from 179 to 189. Would not be better to write these questions at the end of introduction section? I read these questions or suggestions in the middle-paper.
11º. Model building is correct, but authors have to implement other authors which have worked with this metodologhy, so we can compare different opinions, variables, results, etc...
12º. Authors write (Lines 362-365): Demand is a very important factor in the green supply chain network of distributed 362 3D printing intelligent factory. We will analyze the impact of various costs from demand 363 1, demand 10, demand 20, demand 30, demand 40 and demand 50. First of all, the supply 364 chain results of demand 1, demand 20, demand 30 and demand 40 are displayed, as shown 365 in Figure 3. Here, supplier 1 is still selected to supply raw materials. Can authors write and show real examples from some companies, please.
13º. Conclusions section: further analysis if required and authors need to give real examples by companies which has been improved according to their results, please.
11º. Furthermore, authors have to implement Theoretical and practical implications and Limitations and future research subsections, due to the importance of this topic in the industry of what? Logistics, tourism, technology, food.... I do not know it.
Congratulations because I SAW THAT YOU WORKED VERY HARD.
